# Identification and Characterization of Small RNA Markers of Age in the Blow Fly *Cochliomyia macellaria* (Fabricius) (Diptera: Calliphoridae)

**DOI:** 10.3390/insects13100948

**Published:** 2022-10-18

**Authors:** Carl E. Hjelmen, Ye Yuan, Jonathan J. Parrott, Alexander S. McGuane, Satyam P. Srivastav, Amanda C. Purcell, Meaghan L. Pimsler, Sing-Hoi Sze, Aaron M. Tarone

**Affiliations:** 1Department of Biology, Utah Valley University, Orem, UT 84058, USA; 2Department of Entomology, Texas A&M University, College Station, TX 77843, USA; 3Department of Computer Science and Engineering, Texas A&M University, College Station, TX 77843, USA; 4School of Mathematical and Natural Sciences, Arizona State University, Glendale, AZ 85306, USA; 5Department of Molecular Biology and Genetics, Cornell University, Ithaca, NY 14853, USA; 6Centre for Forensic Science, Department of Pure and Applied Chemistry, University of Strathclyde, Glasgow G1 1XQ, UK; 7Department of Biochemistry and Biophysics, Texas A&M University, College Station, TX 77843, USA

**Keywords:** miRNA, Calliphoridae, forensic entomology, development, carrion ecology

## Abstract

**Simple Summary:**

Immature blow fly development information is critical for temporal estimations in death investigations. While there are phenotypic markers of development, they often do not provide adequate precision and resolution. In order to ameliorate these issues, we have investigated microRNAs as putative markers of immature development in the forensically relevant blow fly, *Cochliomyia macellaria*. Using RNA sequencing, we were able to identify miRNAs present in immature developmental stages, as well as identify miRNA which were significantly differentially expressed. Through these analyses, we found likely markers of development time, some of which were validated with qPCR methods. As a follow-up, we investigated factors which may influence expression of miRNA throughout development, such as selected genotypic variation and sex. It was very clear that genetic and abiotic factors can impact predictions of age, some of which may exhibit interactions. It is important that future work investigate the markers most robust to genetic effects on development.

**Abstract:**

Blow fly development is important in decomposition ecology, agriculture, and forensics. Much of the impact of these species is from immature samples, thus knowledge of their development is important to enhance or ameliorate their effects. One application of this information is the estimation of immature insect age to provide temporal information for death investigations. While traditional markers of age such as stage and size are generally accurate, they lack precision in later developmental stages. We used miRNA sequencing to measure miRNA expression, throughout development, of the secondary screwworm, *Cochliomyia macellaria* (Fabricius) (Diptera: Calliphoridae) and identified 217 miRNAs present across the samples. Ten were identified to be significantly differentially expressed in larval samples and seventeen were found to be significantly differentially expressed in intrapuparial samples. Twenty-eight miRNAs were identified to be differentially expressed between sexes. Expression patterns of two miRNAs, *miR-92b* and *bantam*, were qPCR-validated in intrapuparial samples; these and likely food-derived miRNAs appear to be stable markers of age in *C. macellaria*. Our results support the use of miRNAs for developmental markers of age and suggest further investigations across species and under a range of abiotic and biotic conditions.

## 1. Introduction

Decomposition is a critically important ecosystem service that results in the recycling of nutrients from deceased organisms back to the food web. Most research in this realm is on decomposition of autotrophic biomass and feces; however, there is a growing appreciation for the need to consider heterotrophic biomass, including deceased animals, in nutrient recycling [1,2]. Blow flies (Diptera: Calliphoridae) are often considered to be major players in decomposition ecology [3,4]. These flies are primary colonizers of remains, often known to appear within minutes of death [5,6]. Since these flies are adapted to surviving on decaying matter, they have been studied broadly for their importance in forensic entomology [7], cases of myiasis in agriculture [8,9], the spread of microbes [10,11,12], use in maggot therapy in medicine [13,14], and even pollination [15,16,17].

Many of the aforementioned ecological services and applications of blow flies are linked to knowledge of their immature development. In the case of forensic entomology, development information for insects is used in conjunction with knowledge of decomposition ecology to estimate a portion of the post-mortem interval (PMI), or the time since death, associated with insect colonization [18,19]. Traditionally, physical markers, such as size or weight, of each development stage are used to estimate the minimum PMI. While these classical approaches are generally considered to be accurate, the duration of later development stages and lack of easily identifiable intra-stadial markers are limitations to these practices [20]. For example, the third instar larva is the longest larval stadium, yet age–size correlations become increasingly inconsistent [20]. Additionally, there is a high degree of both intra- and inter-species variation in the duration, both absolute and relative, of the feeding and post-feeding larval stages [21,22,23,24]. In some species, such as *Phormia regina* (Meigen), this difference is quite dramatic, where the onset of feeding third instar to intrapuparial development takes up to 13 days, and 8.8 of those days are in a postfeeding stage [22]. Similarly, metamorphosis in the puparium can last for 175 hours at 25 °C, significantly longer than other stages [24]. Therefore, classical approaches to predict fly age based on external physical appearances are imprecise at later stages of development. A variety of destructive and non-destructive attempts have been investigated to improve precision in age estimates with these stadia [25,26,27,28,29,30]. The introduction of molecular markers for age and development would therefore greatly increase the accuracy and precision of the estimation of developmental progress.

Genes (mRNA) important for development can be remarkably conserved across the animal tree of life [31,32], and the patterns by which these genes are expressed, temporally and spatially, drive much of the diversity among species [33,34,35,36]. Transcriptomic information such as this has wide applications across biology, stemming from basic research in development and aging, to the understanding of complex developmental disorders and diseases [37,38]. Though early work in the field of transcriptomics was restricted to model organisms such as *D. melanogaster*, recent advancements in high-throughput sequencing and bioinformatics have opened the door for in-depth genetic analysis of taxa for which prior genomic knowledge was not available.

While there is promise to using mRNA makers for aging immature flies for forensic entomology [29,39,40,41,42,43], the use of mRNAs as markers is also limited by the fragility of mRNAs; mRNAs lack chemical and thermal stability of DNA [44,45] and may break down quickly in the presence of RNases in the environment [46]. An ideal marker of age should be chemically stable and preserved during collection, insensitive to environmental and genetic factors, and can provide time information, especially at later developmental stadia when classical approaches lose precision. One such marker showing promise comprises microRNAs (miRNAs). These small non-coding RNAs are approximately 22 base pairs long in a characteristic double-stranded hairpin structure, which can be conserved across different species [47], making them more robust than mRNAs [48,49,50]. These small RNAs function in RNA silencing and post-transcriptional down-regulation of mRNAs, which can be utilized in the development of the popular RNAi technique [51]. The stability of these small RNAs enables them to be accurately quantified, even in degraded RNA samples, suggesting that they can be useful markers of age and development [48,52,53,54]. Previous studies have shown that while half of these miRNAs are constitutively expressed, many of them demonstrate developmentally regulated expression [48]. The fact that miRNA expression follows a development-dependent manner makes it possible for miRNA to be a marker for insect age [55].

While miRNAs are shown to be differentially expressed (DE) in *Lucilia sericata* (Meigen), a forensically relevant fly, at different developmental points [55], further research into the longer development stadia across species of flies is needed—as is research into other forensically relevant fly species. To address this shortcoming, we performed a preliminary study of small RNA sequencing across the immature development of the secondary screwworm, *Cochliomyia macellaria* (Fabricius). This is the first study, to our knowledge, that utilizes miRNA expression from RNAseq in a developmental context using whole bodies of a forensically relevant dipteran. We hypothesized that there would be an age-associated differential expression of known miRNAs, which will further allow us to identify potential markers of development and group samples by developmental stage. To address how robust these potential markers of age are across a range of intraspecific genotypes, we also performed small RNA sequencing of two different developmental time points of *C. macellaria*, which were selected for fast or slow development time, as well as for a control treatment. Subsequently, additional sequencing was performed to compare samples by sex. We hypothesized that using expression information from wild-type development will allow us to train a random forest model to predict both stage and development percentages of samples from development time selection experiments. After assessing miRNA from RNAseq data, we also worked to validate these markers with qPCR, preliminarily identifying two miRNAs that appear to perform as expected in novel samples. Additionally, we identified a number of miRNA from the insects’ vertebrate food source, which can be useful in differentiating feeding vs. postfeeding stages.

## 2. Materials and Methods

### 2.1. Insect Maintenance and Collection

In July of 2015, wild-type adults of *Cochliomyia macellaria* were collected from carcasses in College Station, TX, USA (30°36′05″ N 96°18′52″ W) and identified morphologically. Voucher specimens were deposited in the Texas A&M University Insect Collection under the voucher numbers 728 (selection lines) and 731 (wild-type). Blow flies were maintained in a BugDorm plastic cage (BioQuip 1452 Bug Dorm, 11.75” cube) and provided with fresh deionized water and refined sugar ab libitum. Fresh bovine blood was provided as a protein source for oogenesis on alternating days. Colonies were maintained at 28 °C for a 14:10 light:dark (L:D) photoperiod. Four replicate cohorts of approximately 200–250 embryos were collected and raised on 100 g of bovine liver in 1.1 L glass canning jar with approximately 100 g of sand. Jars were placed in a growth chamber (136LLVL Percival, Percival Scientific Inc.) and reared at 25 °C with 50% humidity on a 14:10 L:D cycle. Growth chambers were calibrated before the initiation of the study and checked daily to ensure that temperatures were maintained. Samples often referred to as “puparial” or “pupal” will be referred to as “intrapuparial” in this manuscript, though it should be noted that pupal and intrapuparial periods of development differ [56]. Samples for wild-type *C. macellaria* from the following life history “stages” were flash-frozen and stored at −80 °C until RNA isolation: feeding third instar, early and late postfeeding 3rd instar larvae, early intrapuparial, early–mid intrapuparial (mid intrapuparial 1), mid–late intrapuparial (mid intrapuparial 2), and late-stage intrapuparial. Additionally, *C. macellaria* collected in Longview, TX in July 2013, selected for 13 generations for fast and slow development, along with a non-selected control treatment from the same collection event, were collected at the early postfeeding and mid–late intrapuparial time points to compare the potential range of expression across genotypes (collection and development times in Table 1). Whole jars of larvae or intrapuparial samples were sampled, while control jars were used for comparison of development time. The selection response for these strains can be seen in [57] and [58].

### 2.2. Small RNA Isolation and Sequencing

RNA was extracted from individuals using a Qiagen miRNeasy Kit and on-column DNase treatment following manufacturer protocols (Qiagen Inc., Valencia, CA, USA) and then was stored with Superase In RNase inhibitor at −80 °C. Samples were not identified as to their sex, as there is not known to be significant differences in the development time between sexes of *C. macellaria* [58]. Sample concentration and quality control were assessed with a Nanodrop Spectrophotometer (Thermo Fisher Scientific Inc., Wilmington, DE, USA). Samples were sequenced following standard protocols for Illumina TruSeq small RNA library construction, 2S rRNA depletion (following Blenkiron, Tsai [55]), and sequencing at the Texas A&M University AgriLife Genomics and Bioinformatics Services facility on rapid mode of an Illumina HiSeq 2500. Sequencing quality control ensured that each sample had approximately three million reads, each about 50 bp long [59,60]. For each time-point and selection treatment, three individual samples were sequenced (biological replicates) twice each (technical replicate), resulting in 78 libraries, with six libraries for each time-point/treatment. Short RNA reads were deposited in SRA (selection lines: PRJNA548589, wild-type: PRJNA548590).

In order to estimate effect of sex, *C. macellaria* were reared as above and identified as male or female using a recently optimized mRNA method as in [61]. Five male samples and six female samples were extracted as above and sequenced by Scripps Research with five million reads per sample on the Illumina NextSeq 500 platform (San Diego, CA, USA). Short RNA reads were deposited in SRA (sex identified: PRJNA876743).

### 2.3. Sequencing Data Analysis

All data parsing through sequence identification were completed using the Brazos Cluster (Brazos Computational Resource, Academy for Advanced Telecommunications and Learning Technologies, Texas A&M University). Scripts for this processing can be located at https://github.com/cehjelmen/cmac-miRNA (accessed on 8 September 2022). Adaptors were trimmed using cutadapt-1.10 and filtered to include all reads between 20 and 25 nucleotides in length. Unique numbers of reads for each length between 20–25 nucleotides suggested we adequately sequenced miRNA for each sample (Table 1) A unique sequence database was generated from sequence reads and all unique sequences were mapped to Flybase and all miRNAs in miRBase to identify conserved miRNAs [60,62,63,64,65,66]. Counts of each miRNA were output to a .csv file to be further parsed for sequence analyses. To be included for further statistical analyses, each miRNA had to be present with greater than ten counts. The miRNA expression levels were normalized and differential expression at different development time points was analyzed using DESeq2 in R [67,68]. Both fold-change values and adjusted *p*-values were calculated for normalized data (Appendix A). Significantly differentially expressed (SDE) miRNAs were visualized in heatmap format. Comparisons of miRNA expression level were also made between the early postfeeding (EPF) and mid–late intrapuparial time points (MP2) for lines of *C. macellaria* selected for development time to determine which miRNA markers are indicators of genotypic differences in development. R scripts used for analyses can be found at https://github.com/cehjelmen/cmac-miRNA (accessed on 8 September 2022).

Normalized expression data for each sample in each development time point were further investigated with a Principal Component Analysis (PCA) using the prcomp() command in R, and visualized with the ggplot2 and ggbiplot packages [69,70]. Normalized expression data for selection lines were then plotted on the PCA for wild-type development for comparison across genotypes. Random forest machine learning model analyses were performed on normalized expression data using the randomForest() function in the randomForest package in R in order to identify which miRNAs contribute most to the correct identification of development time points in wild-type *C. macellaria* [68,71]. Models were built with 5000 trees using both percentage data and development time point data. Additionally, normalized expression data for wild-type *C. macellaria* were used to train the random forest model to predict development time points and the percentage of development for flies selected for development time.

### 2.4. Evaluation and Validation

Sixteen miRNAs were selected for validation through qPCR based on previous knowledge on development and information on SDE miRNAs in development of *C. macellaria* (invertebrate: *miR-1, miR-8, miR-10, miR-31a, miR-92b, miR-184, miR-276b, miR-277, miR-305, miR-317, miR-957, miR-bantam*, and *miR-let7*; vertebrate: *miR-10a* and *miR-22*). Intrapuparial samples were selected for validation with qPCR, as intrapuparial development accounts for most of the development time and cannot be morphologically separated as clearly as larval development. Total RNA was extracted using an miRNeasy Kit (Qiagen, Hilden, Germany) from three individual pupae for three independent biological replicates of four distinct pupal time points each. Intrapuparial samples were used for qPCR as intrapuparial development accounts for the largest portion of development. RNA quality and quantity were assessed using a DeNovix spectrophotometer DS-11. Then, 1µg of total RNA was treated with DNase I Amplification grade (Invitrogen, Thermo Fisher Scientific, Carlsbad, CA, USA) and was reverse transcribed using the miScript RT II kit (Qiagen, Hilden, Germany) in HiFlex buffer. Similarly, a negative RT reaction was set up to test for DNA contamination. Quantitative PCR using the miScript SYBR green kit were conducted on a Bio-RAd CFX96™ Real-Time PCR Detection System (Bio-Rad Laboratories, Hercules, CA, USA). In total, 20 µL reactions in duplicates were set up with 2 ng of cDNA and 0.4 µM small RNA specific (forward) primer and 2 µL of miScript universal (reverse) primer. Reaction conditions consisted of an initial activation step of 95 °C for 15 min, and 40 cycles of denaturation of 94 °C for 15 s, annealing at 55 °C for 30 s, and extension at 70 °C for 30 s. Geometric means of 2S rRNA and U6-snRNA expression were used to normalize the target miRNA expression levels [72]. Fold-change values were calculated using the 2(−∆∆Ct) method [73].

## 3. Results

### 3.1. Sequencing Output and Identification of miRNA

Small RNA sequencing resulted in approximately 9,000,000–16,000,000 reads per *C. macellaria* sample (Appendix A), of which approximately 700,000–3,500,000 were unique sequences (Table 1). While we focused on reads between 20–25 nucleotides, there was consistently a large peak of reads present in the distribution from 26–31 nucleotides, likely indicating piwi-interacting RNAs (piRNAs). The proportion of reads from 26–31 nucleotides increased throughout development, both in terms of the entire sample and its relation to miRNA reads (Appendix A). After filtering reads to lengths of 20–25 nucleotides and mapping to miRbase and Flybase miRNA databases (Distribution of read counts by read length found in Appendix A), we were able to identify 217 different miRNAs in our samples of *C. macellaria* (Appendix A). Raw counts of each miRNA for each sample can be found in Appendix A. When compared to other miRNA studies in calliphorid and muscid flies, 115 mapped miRNAs were found to be unique to our study (Appendix A, Appendix A).

### 3.2. Summary of Results for All of Immature Development

A principal component analysis (PCA) with normalized expression data for all miRNAs identified in this study could differentiate the development time points in wild-type flies reliably, apart from feeding third instar from early postfeeding third instar (39.2% of variation between PC1 and PC2, Figure 1).

A random forest analysis using all normalized miRNA expression information from wild-type *C. macellaria* predicted the development time point of samples with an out-of-box error rate of 19.05% (Appendix A). These errors in placement were always with adjoining time points (e.g., early postfeeding classified as feeding third instar, late postfeeding classified as early postfeeding, etc.) (Appendix A). When percentage development data were used to train the model, 86.62% of variation was explained.

### 3.3. Identification of Vertebrate miRNA

Out of the 217 miRNAs identified, 58 were identified from vertebrates and may represent a signal from the larval food source (bovine liver) (Figure 1 and Appendix A). Removal of miRNA sequences identified as vertebrate from the PCA of wild-type *C. macellaria* had a modest reduction in the percentage variation explained when compared to using all miRNA data, and could separate feeding third instar and early postfeeding third instar (38.8%, Figure 1A,B). These results are not surprising given that the immatures were provided with liver from *Bos taurus* as a food source. If all miRNA sequences matched to *Bos taurus* were plotted across development time, there is a sharp drop to zero expression with late postfeeding third instar larvae, suggesting that the identification of *B. taurus* miRNA may be helpful in distinguishing early and late postfeeding third instar larvae (Figure 1).

### 3.4. Analysis of Larval and Intrapuparial Samples with PCA and DESeq2 Methods

As immature flies are easily visually differentiated as either larval or intrapuparial samples, further analyses were run separately on larval stages and intrapuparial stages. DESeq2 analysis found ten SDE miRNAs among wild-type larval samples and 17 among wild-type intrapuparial samples (adjusted *p* < 0.05, log2fold change >−1, Appendix A, Figure 2A,B, Appendix A. A PCA of these SDE miRNA successfully separated samples by development point and could explain 87.5% and 79.3% of the variation within the first two components for larval and intrapuparial samples, respectively (Figure 3A,B). In samples from *C. macellaria* selected for development time, DESeq2 analysis found 39 SDE miRNAs among treatments in early postfeeding third instar larvae, four of which were also identified to be SDE in wild-type larval samples (Figure 2C and Appendix A, Appendix A).

In mid-intrapuparial samples from development time selected flies, 54 miRNAs were identified as SDE among treatments, 13 of which were identified as SDE in wild-type intrapuparial samples (Figure 2D and Appendix A, Appendix A). Among the SDE miRNAs in development time selected flies, 18 were identified in both sampled time points (Appendix A). When normalized expression information from early postfeeding third instar development time selected samples was projected onto the PCA from wild-type larval development, they clustered near the wild-type early postfeeding third instar samples (Figure 3A). In the case of the development time selected for mid-intrapuparial samples projected onto the wild-type intrapuparial PCA, fast and control samples clustered near mid-intrapuparial 1, which have similar development percentages (74.4–75.0% in selected vs. 70.0% in wild-type samples) (Figure 3B). Slow mid-intrapuparial samples clustered closely with the wild-type mid-intrapuparial 2 samples (74.7% in slow vs. 90.5% in wild-type) (Figure 3B).

### 3.5. Random Forest Models and Predication of Development Time

Random forest regression models were trained for both larval and intrapuparial samples using normalized expression information from wild-type samples for miRNA identified to be SDE in wild-type samples. The larval model could explain 68.06% of the variation (mean of squared residuals = 0.0016), while the intrapuparial model could explain 84.59% of the variation (mean of squared residuals = 0.0056). When used to predict the development proportion of development-time-selected samples, the larval random forest model predicted the development proportion with high accuracy (within 1.2% of correct development proportion, Figure 4A, Appendix A). In the case of intrapuparial development-time-selected samples, there was less accuracy (within 16.4% of correct development proportion). The intrapuparial random forest model under-predicted the proportion of development for fast and control samples by about 10% and over-predicted the proportion of development in slow-development-selected samples by about 14% (Figure 4A, Appendix A). The predictions of development proportion for intrapuparial samples mirrors the clustering pattern of the PCA projections, as slow-developing-samples clustered closely with wild-type samples later in intrapuparial development (Figure 3B).

### 3.6. qPCR Validiation

Validation with qPCR found two miRNA that exhibited significantly different levels of expression among intrapuparial development time points, *miR-92b* (Figure 5A,B) and *bantam* (Figure 5C,D). Other miRNAs, such as *miR-957*, had high support for use according to random forest and DESeq2 analyses, but were not validated with qPCR (Figure 5E,F). When investigating *miR-92b*, both qPCR and transcriptomics found higher expression in earlier intrapuparial development, which was even higher in larval development (Figure 5). As for *bantam*, both miRNA transcriptome evidence and validation through qPCR found higher expression in the late intrapuparial time point, with low expression in earlier development (Figure 5). However, in the case of *miR-957*, there was higher expression with subsequent development stages, suggesting that greater sampling is required to clarify the points in development defining the period of time where expression increases. The two miRNAs validated in intrapuparial samples (*miR-92b* and *bantam*) were used in a random forest regression model to predict the development proportion of intrapuparial samples as above. Using this model, 52.36% of the variation was explained (mean of squared residuals = 0.0172). When using the random forest regression model trained with miRNA identified as potential markers with qPCR validation, samples were predicted to have completed a lower proportion of development than had occurred (Fast-, Slow-, and Control-developed samples, Figure 4B). While there was an under-prediction in development proportion, there did not seem to be an effect of selection regime, suggesting these markers are more robust to the genotypic variation in these selected lines, suggesting that more markers may be needed for a robust prediction of age with this random forest model (Figure 4B).

### 3.7. Analyses of Sex Biased Expression of miRNA

Subsequent sequencing was performed on early intrapuparial samples in which sex was identified with mRNA methods [61]. These new miRNA reads were mapped to the 217 miRNAs identified in the initial portion of this study (Appendix A). When comparing between sexes in early intrapuparial samples reared at 25 °C, we found that 28 miRNAs were SDE by sex (Appendix A). Of these 28 SDE miRNAs, only three overlapped with those identified as SDEs in wild-type intrapuparial samples (*miR-305, mir-375*, and *miR-957*), two were identified to be SDE in wild-type larval samples (*miR-305* and *miR-317*), and only one was shared between all three (*miR-305*) (Appendix A, Appendix A). While five of these SDE miRNAs were ones selected for validation with qPCR, there were no differences in expression by sex for either *bantam* or *miR-92b*.

## 4. Discussion

To determine if there was age-dependent miRNA expression in the blow-fly *Cochliomyia macellaria*, we sequenced, identified, and analyzed the expression of known miRNAs across seven developmental time points in immature development (three third instars, four intrapuparial) of wild-type colony-raised flies. While miRNA have been identified in other flies of similar forensic, medical, and veterinary relevance [55,74,75], this is the first study outside of *Drosophila*, to our knowledge, which investigated patterns of miRNA expression using RNAseq in the whole insect in a developmental context. Our methods allowed us to identify 217 distinct, known animal miRNA sequences in *C. macellaria* (Appendix A). Of these 217 miRNAs, 58 were mapped to known vertebrate miRNAs and had higher prevalence in feeding third instar and early postfeeding larvae, suggesting they are a signal from the larval food source (Appendix A). Although many of these sequences have been found in related species of flies, our sequencing depth and sampling across development allowed us to identify 115 miRNAs not identified in the aforementioned studies on medically and forensically relevant flies (Appendix A). As this work is a preliminary study on the expression of miRNA in the whole body of flies across developmental time, we only investigated and identified previously known miRNA sequences from FlyBase and miRbase. There are likely novel miRNAs to be discovered from our sequencing files, which could be identified in a future study.

In wild-type samples, most development points clustered distinctly in a PCA using normalized expression values for all identified miRNA, apart from some overlap in feeding third instar and early postfeeding third instar development points (Figure 3). This overlap could be because they are both within the third instar but could also be due to residual miRNAs from the vertebrate food source rather than those endogenously produced by the maggots. When plotting normalized counts of the sequences mapped to *Bos taurus* miRNA (*miR-191, miR-21, miR-22, miR-2904, miR-30a*), we found a high detection in feeding third instar larvae, as well as early postfeeding third instars, with a sharp drop to zero detection in the late postfeeding and intrapuparial samples (Figure 1C and Appendix A). Interestingly, when the 58 vertebrate miRNAs are excluded, we can clearly separate the feeding third instar and early postfeeding larvae in our PCA analyses with only a slight difference in the variance explained by the first two principal components (Appendix A). This information may help differentiate feeding and postfeeding third instar larvae in forensic entomology, which has been an acknowledged challenge in the field [20,22].

The presence of vertebrate miRNA may also be useful in identifying if evidentiary samples were feeding as expected, since some cases have situations in which the larval feeding substrate is ambiguous without molecular identification (e.g., a body found in a dumpster that contains discarded food of animal origin). While cytochrome b sequencing has been used to confirm the source of larval meals in the past [76,77,78], miRNAs would simultaneously provide information regarding larval gut content and information that could more precisely identify larval age. However, due to similarities in miRNA between vertebrates, this information may not be able to differentiate the source of the gut content as reliably as mitochondrial DNA. Nevertheless, vertebrate miRNAs in larvae may simultaneously help identify if the insects were feeding as expected and aid in visual cues to help separate older postfeeding third instars from younger ones. Additionally, the increasing presence of piRNA relative to miRNA in samples may provide a useful tool for differentiating immature states but will require more investigation (Appendix A).

We found that miRNA expression in whole individuals differs significantly across development time for *C. macellaria* and can indicate the proportion of development completed according to three distinct analytical approaches: PCA, DESeq2, and random forest. The ability to predict development proportion is not surprising, given that miRNA expression is known to play a role in mRNA regulation [79], which is known to have a relationship with development in forensically relevant flies [39,40,42]. The SDE miRNAs identified in our study have been shown to be involved in developmental pathways in *Cochliomyia hominivorax* (Coquerel), *Lucilia sericata* (Meigen), and *Stomoxys calcitrans* (Linneaus) (Appendix A). For example, the SDE miRNAs from our study were found to be involved in the regulation of pathways involved in metabolism [80,81,82], neural development and maintenance [83,84,85], as well as circadian pathways [86]. The ability of miRNA expression to differentiate developmental time points and their structural stability compared to mRNA, make them ideal candidates for markers of development time and justify further studies within *C. macellaria* and across other species of forensic, medical, and veterinary importance.

Previous work on this fly has found developmental differences between different populations and between years of collection [58,87]. Therefore, to address the consistency of the observed expression patterns across a range of genotypes and populations, we performed small RNAseq on two development time points in three populations of *C. macellaria*, two selected for fast or slow development and their unselected control. When comparing among selection lines, we found the fast and control lines have very similar expression patterns, but differ significantly from the slow development lines, suggesting that selection has led to divergence in miRNA expression (Figure 2C,D). The DE miRNAs between these developmentally selected samples have been shown to be involved in development, immunity, nervous system, and metabolic pathways (Appendix A). For example, in the fast and control lines, there was a higher expression of *miR-305* than the slow development lines in both larval and intrapuparial samples (Appendix A), a miRNA known to be involved with the insulin-signaling and Notch pathways in *Drosophila*, as well as adaptation to nutrient deprivation [88,89]. These results suggest that these selection lines have not only diverged significantly in their phenotypes [57,58], but also in their genotypes and associated gene expression patterns, raising the possibility that some miRNAs identified here represent the difference between chronological and physiological/biological age in intrapuparial samples [90,91,92,93].

We found that models based on expression information from SDE miRNA in wild-type samples could estimate the proportion of development in samples selected for development time with reasonable accuracy (Figure 2 and Figure 3), supporting the hypothesis that variation in miRNA expression may be a robust predictor of the proportion of development across genotypes in *C. macellaria*. In both PCA and random forest models, the proportion of development completed for development-time-selected samples was accurately predicted in larval samples (Figure 2A and Figure 3A, Appendix A). Within intrapuparial samples, the proportion of development was slightly underestimated in fast-developing and control development samples, whereas the proportion of development in slow samples was overestimated (Figure 2B and Figure 3A, Appendix A). As there is a difference between chronological age and physiological/biological age, it is possible that selection for development time in *C. macellaria* acted on genes related to temporal age rather than physiological age [92,93,94]. If this is the case, slow-developing flies, while developing for the same proportion of development as fast and control flies, have spent more time in development, which, in turn, could lead to overestimation of the proportion of development. This issue of temporal vs. physiological age may be exacerbated in the intrapuparial stage, as it is the longest stadium of development. Overestimation and underestimation of development proportion in genotypically variable samples, as well as their clustering with either an early–mid-intrapuparial or late intrapuparial development, suggest that we may be able to at least partition the longest stadium into three portions: early, mid, and late. The differentiation of slow-developing flies, however, does suggest an upper limit to variation in estimates due to genetic differences among individuals.

Two miRNAs validated with qPCR were found to have significant differences in expression between developmental time points (*miR-92b* and *bantam*, Figure 5A–D). Based on previous work in *Drosophila*, these two miRNAs likely have important roles in intrapuparial development in *C. macellaria*. In *Drosophila*, *miR-92b* is involved in regulating *myocyte enhancer factor-2* (*Mef2*) levels during muscle development [95]; while *bantam* has been known to be involved in growth, cell proliferation and proapoptotic genes [96,97]. While the patterns of expression hold for the SDE miRNA validated with qPCR, the scales of response seen in the qPCR analyses and small RNAseq data are not equivalent. The reduced scale of response seen in qPCR analyses suggests that these differences in miRNA expression across time may be subtle and difficult to validate with a low sample size. For example, *miR-957* was identified to be a probable marker of development proportion from both preliminary DESeq2 and random forest analyses, yet qPCR could not validate the expression pattern (Figure 5E,F). Selection of miRNAs for validation was also performed before the identification of the miRNAs, which were SDEs among selection genotypes and sexes, leading to the selection of several miRNAs which are now known to vary based on genotype or sex. For example, *miR-957* was found to be DE between sexes in follow-up sequencing, suggesting variation in expression due to differences in sexual development (Appendix A). Increasing the sample number in qPCR, as well as the careful selection of the future miRNA of interest, will likely increase number of putative markers for the proportion of development in *C. macellaria*. While we have identified and preliminarily validated two potential miRNA markers of development in *C. macellaria*, future work should validate the expression of these markers in a variety of environmental conditions to ensure their reproducibility and robustness.

Further work must be done to further validate this miRNA dataset to identify useful markers for estimating the proportion of development in *C. macellaria*. For example, identified markers should be used to estimate the proportion of development in known samples compared to the standard thermal summation modeling. This further validation would provide information on if these miRNA markers reduce error in estimation when compared to traditional methods. As molecular methods are more expensive and require more energy, it is necessary for future work to address this validation. In addition to further validation with qPCR, other known sources of variation in development should be investigated to ensure the markers identified are robust to environmental changes and genotypic differences. One such factor which results in variation in development time is sex. Phenotypically, there is little difference in development progress between males and females in *C. macellaria* [58], so we did not separate sexes in the initial sequencing for this manuscript. However, a lack of phenotypic differentiation does not guarantee a lack of molecular (e.g., miRNA) differentiation between the sexes. Therefore, our subsequent sequencing efforts used recently optimized mRNA expression identification of sex [61] to determine sex-specific differences in miRNA expression at one developmental time-point. While 28 miRNAs were identified as DE by sex at the early intrapuparial stage, there was very little overlap (three miRNA) of these identified miRNAs with those found to be DE in wild-type intrapuparial development, none of which represented the two qPCR-validated ones (Appendix A). While there were not many shared DE miRNAs by sex and by wild-type development, there were more shared between those by sex and by development-time-selected intrapuparial samples (13 miRNA) (Appendix A, Appendix A). When comparing the intersection of development-time-selected intrapuparial samples, wild-type development of intrapuparial samples, and sex, only two miRNAs overlap (Appendix A, Appendix A). These results suggest that future work should incorporate analyses of miRNA expression diagnostic of sex across more developmental time-points [61,98]. As we reared our samples at a consistent temperature, humidity, and light:dark cycle, it is also unknown how abiotic factors such as light or temperature, which are known to affect the development of flies, impact patterns of miRNA expression across development time. Further, it is unknown how consistent these patterns of miRNA expression are across species within the family Calliphoridae and within other insect families of forensic, medical, and veterinary importance.

## 5. Conclusions

In conclusion, we have furthered the understanding of *C. macellaria* development with respect to miRNA expression with the goal of use in forensic sciences. We identified additional miRNAs not previously identified in the species, profiled their expression across larval and intrapuparial development in wild-type and development-time-selected flies, demonstrated the ability to predict ages of development-time-selected samples to within 16.4% of total development, and have validated some of these with qPCR. In addition, we identified the potential to define larval diet by the vertebrate miRNAs found in their guts and showed that this signal can be used to molecularly diagnose feeding versus postfeeding third instar larvae. In doing so, we provide a basis for additional molecular and genomic studies of blow fly development and the ecological, medical, and applied problems with which they are associated.

## Figures and Tables

**Figure 1 insects-13-00948-f001:**
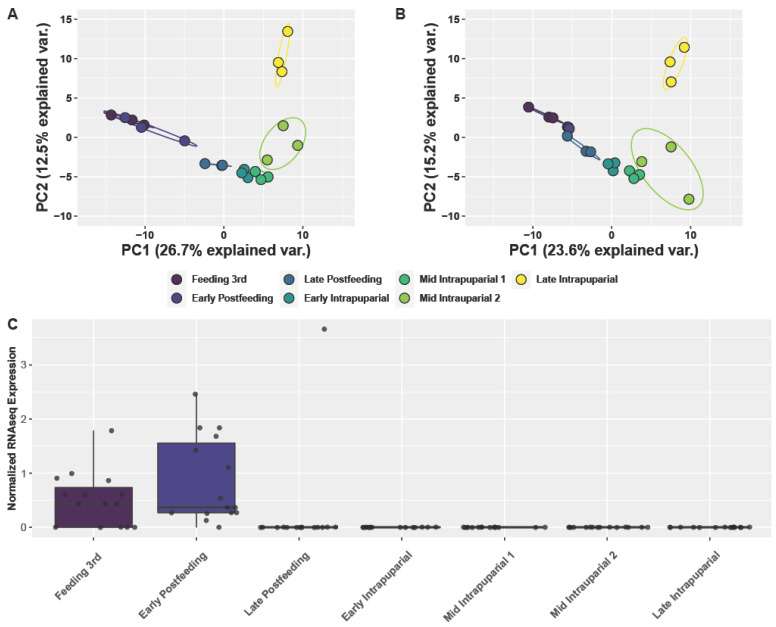
Identification and impact of vertebrate miRNA on the prediction of the development of immature *C. macellaria.* (**A**) PCA of normalized expression information for all identified miRNAs across development of *C. macellaria*. (**B**) PCA of normalized expression information for all identified miRNA across development of *C. macellaria*, excluding those which were matched to vertebrates from the database. Removing these vertebrate sequences had a modest effect on the percent variance explained, but was able to separate the feeding 3rd instar and early postfeeding 3rd instar larval samples. (**C**) Boxplot with jitterplot of miRNA levels for miRNA identified to be from *Bos taurus*, the provided food source, across immature developmental stages. Expression level drops to nothing after the early postfeeding 3rd instar stage.

**Figure 2 insects-13-00948-f002:**
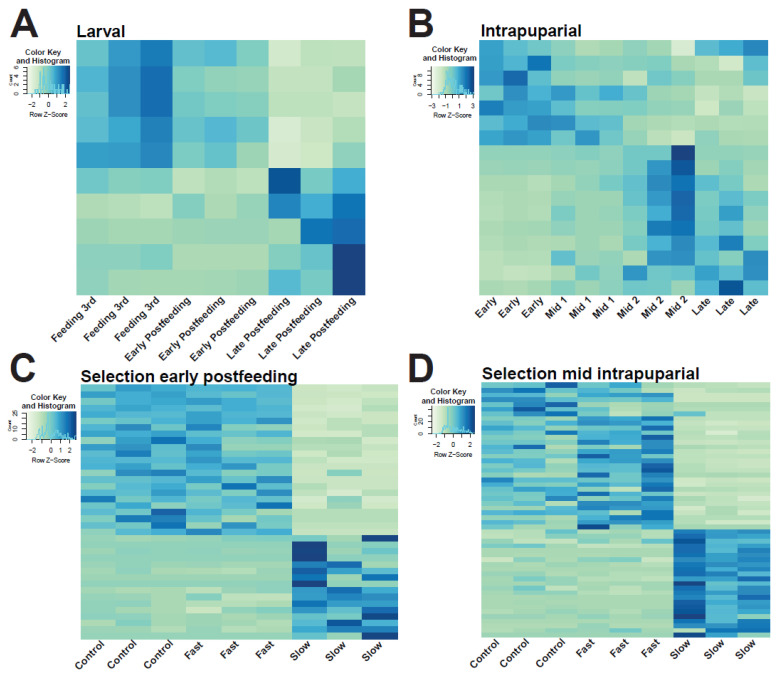
Heatmaps of significantly differentially expressed (SDE) miRNAs from DESeq2 Analyses. (**A**) SDE miRNA across larval development in wild-type *C. macellaria*. Names, fold-change, and *p*-values can be found in Appendix A. Heatmap with miRNA labels can be found in Appendix A. (**B**) SDE miRNA across intrapuparial development in wild-type *C. macellaria*. Names, fold-change, and *p*-values can be found in Appendix A. Heatmap with miRNA labels can be found in Appendix A. (**C**) SDE miRNA across development time selected early postfeeding larval samples of *C. macellaria*. Names, fold-change, and *p*-values can be found in Appendix A. Heatmap with miRNA labels can be found in Appendix A (**D**) SDE miRNA across developmental time in terms of mid-intrapuparial samples of *C. macellaria*. Names, fold-change, and *p*-values can be found in Appendix A. Heatmap with miRNA labels can be found in Appendix A. Darker colors indicate higher levels of expression.

**Figure 3 insects-13-00948-f003:**
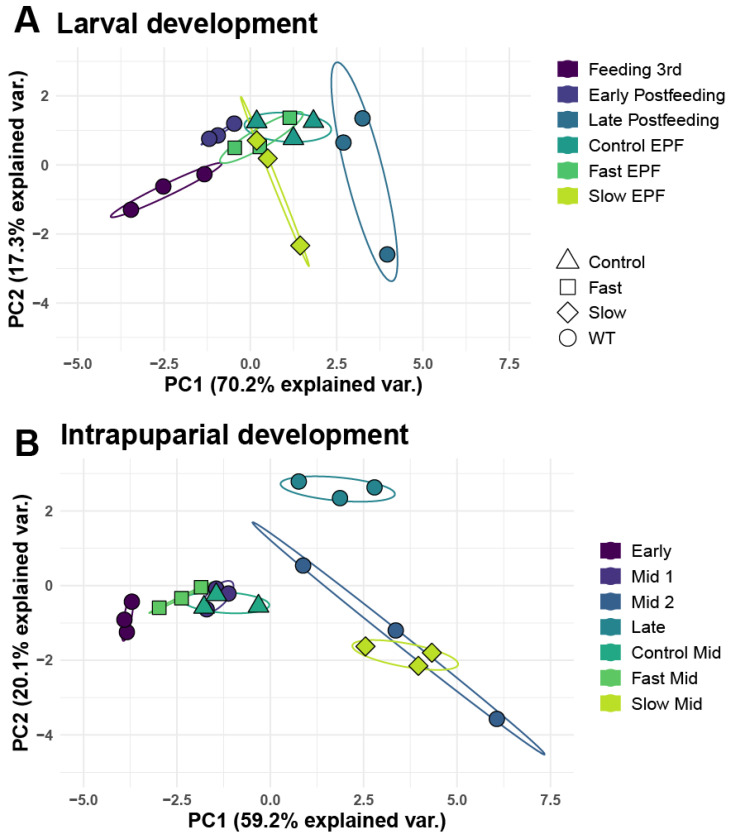
PCA of significantly differentially expressed (SDE) miRNA across development of *C. macellaria.* PCA of normalized expression of miRNA identified to be SDE in wild-type samples (circles) (**A**) PCA of larval development time points could explain 87.5% of the variance between the first two principal components. (**B**) PCA of intrapuparial development time points could explain 79.3% of the variance between the first two principal components. Samples selected for fast development time (square), slow development time (diamond), and their controls (triangle) were projected onto the PCA from the wild-type samples to determine clustering position of new samples from the PCA model built from SDE miRNA from wild-type samples.

**Figure 4 insects-13-00948-f004:**
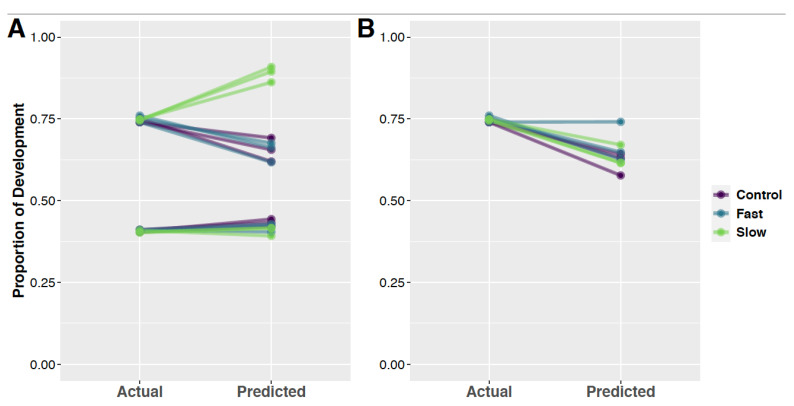
Prediction of the proportion of development completed of development-time-selected samples using a random forest model with all data (**A**) and with two validated miRNA (**B**). (**A**) Random forest models were trained with normalized expression information for miRNA determined to be significantly differentially expressed in larval and intrapuparial development in wild-type *C. macellaria*. Normalized expression information from larval (bottom) and intrapuparial (top) samples selected for differential development times and their controls were inputted into the model to predict the proportion of development completed. The actual proportion of development completed for each sample is on the left, while the proportion of development predicted to be complete from the random forest model is on the right. Larval samples were predicted with high accuracy. The proportion of development for fast and control intrapuparial samples was slightly underestimated, while the proportion of development for slow-developing samples was overestimated. Actual proportions of development and predictions can be found in Appendix A. (**B**) Random forest models were trained on normalized expression of *miR-92b* and *bantam* from wild-type samples. This model was used to predict the development time of development-time-selected samples (Fast, Slow, and Control). This model tended to under-predict the development time of the development-time-selected samples by about 10–15 points. While there was an under-prediction, there did not seem to be a large effect of the selection regime, suggesting these markers are more robust to the genotypic variation in these selected lines. These results suggest that fast-developing lines were more accurately predicted.

**Figure 5 insects-13-00948-f005:**
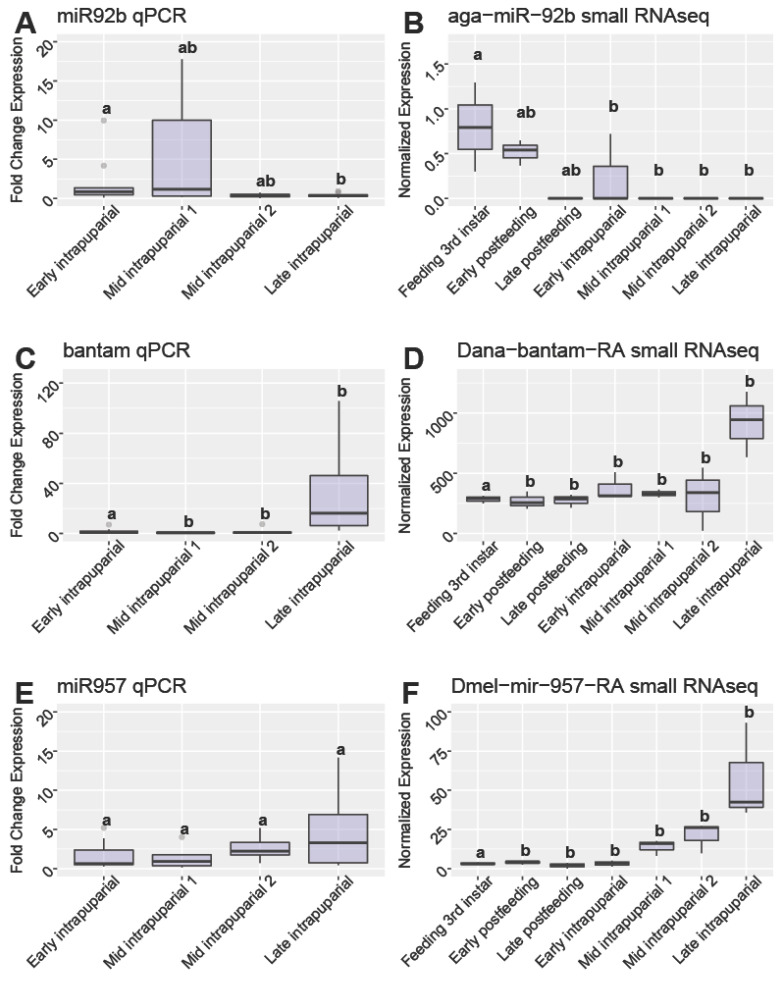
Comparison of expression qPCR and RNAseq. Expression of qPCR (fold-change expression, left) and RNAseq methods (normalized expression) (right) Letters correspond to results of Tukey’s HSD posthoc test. Different letters between groups within plots indicate significant differences (*p* < 0.05). Shared letters indicate no significant difference between groups. (**A**) *miR-92b* qPCR and (**B**) *miR-92b* RNAseq methods. Expression in qPCR samples was found to be significantly higher in mid-intrapuparial 1 than in late intrapuparial samples according to Tukey’s HSD (*p* < 0.05). (**C**) *bantam* qPCR and (**D**) *bantam* RNAseq methods. Expression in qPCR samples was found to be significantly higher in the late intrapuparial development point than the early intrapuparial samples according to Tukey’s HSD (*p* < 0.05). (**E**) *miR-957* qPCR and (**F**) *miR-957* RNAseq methods. Expression was not found to differ significantly between development time points in qPCR samples for *miR-957*, according to Tukey’s HSD (*p* < 0.05).

**Table 1 insects-13-00948-t001:** Collection and read counts for each *C. macellaria* sample. Number of Total Reads and Unique Reads for each development point for reads up to 50 nt. Hours for development listed, as well as the percentage of development complete. Numbers for each sample is the sum of the two technical replicates for all three biological replicates (six libraries for each time-point). Distribution of total read counts across read length can be found in Appendix A. Wild-type *C. macellaria* had an average total development time of 265 h (ranges 234–294 h). Control, and fast- and slow-developing lines collected from Longview, Texas, USA had average development times of 285 (270–293 h), 255 (225–263 h), and 312 (299–334 h) hours, respectively. When the percentage is 100% for late intrapuparial samples, this indicates that some adults had emerged when the samples were collected.

Samples (Three Bio-Reps, Two Technical Reps)	Hours to Collection (Development % Complete)	Total Reads (All Lengths)	Unique Reads (All lengths)	Total Reads Counts (20–25 nt)	Unique Reads (20–25 nt)	Counts of Reads for Predicted miRNA
Feeding 3rd instar	76–80 h (29.9%)	13,210,010	2,858,189	3,513,336	767,584	220,921
Early postfeeding	102–106 h (40.5%)	14,952,640	3,213,766	3,657,951	873,629	218,698
Late postfeeding	120–124 h (46.9%)	10,088,276	944,756	268,403	100,713	23,902
Early intrapuparial	136–140 h (52.2%)	8,887,998	697,479	237,896	74,940	32,738
Mid-intrapuparial 1	180–184 h (70.0%)	10,959,838	1,467,170	578,729	152,024	36,722
Mid-intrapuparial 2	236–240 h (90.5%)	10,731,968	1,134,567	348,102	93,491	15,441
Late intrapuparial	260–264 h (100.0%)	10,368,840	1,159,427	209,517	89,137	20,930
Control early postfeeding	115–119 h (40.7%)	15,604,997	3,148,131	1,819,746	593,846	147,927
Control-mid-intrapuparial 2	212–216 h (74.4%)	15,912,519	2,551,141	1,150,734	359,491	100,406
Fast early postfeeding	104–108 h (40.9%)	16,167,408	3,530,944	2,426,556	719,518	190,589
Fast mid-intrapuparial 2	192–196 h (75.0%)	14,619,401	3,005,893	2,103,762	585,688	139,318
Slow early postfeeding	127–131 h (40.5%)	10,596,116	1,221,303	449,203	144,935	9,295
Slow mid-intrapuparial 2	233–237 h (74.7%)	14,027,331	1,762,262	1,424,884	217,608	45,952

## Data Availability

Voucher specimens were deposited in the Texas A&M University Insect Collection under the voucher numbers 728 (selection lines) and 731 (wild-type). Raw sequences files can be found at NCBI SRA under the following project numbers: selection lines: PRJNA548589, wild-type: PRJNA548590, sex identified: PRJNA876743. Scripts for use on a supercomputer as well as R scripts for analyses can be found at https://github.com/cehjelmen/cmac-miRNA (accessed 8 September 2022).

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
