# Peer review of "Identification and Characterization of Small RNA Markers of Age in the Blow Fly Cochliomyia macellaria (Fabricius) (Diptera: Calliphoridae)"

_insects, 2022, doi:10.3390/insects13100948_

Round 1

Reviewer 1 Report

Major Comments:

This is a very thorough and well-executed study investigating the possibility of using microRNAs to age forensically relevant insects. Specifically, the authors focused on the post-feeding third instar larval stages and various timepoints throughout the intrapuparial period, which are all notoriously difficult to accurately estimate using thermal summation models. This work is the culmination of a massive amount of lab experiments and molecular analyses, all of which resulted in some very interesting and useful findings. The authors have also outlined additional future experimentation and analyses that they see as being integral to fully understanding the application of microRNAs in forensic entomology.

A major concern with this and other cutting-edge research concerning novel methods for age estimation is the expense and energy that would need to be invested by an investigating agency to get results that might be only slightly more helpful than using the standard methods (that are much cheaper) that have been around for decades. Don't get me wrong, the research is amazing, but the application to a real death investigation is uncertain. It would greatly strengthen the applied potential of this paper to have a direct comparison of this method to standard estimation methods. How well does the microRNA method hold up to thermal summation modeling? At the very least, a discussion of the importance of such validation work in future studies would be appreciated.

Minor Comments:

Title and Abstract: The author name (Fabricius) should directly follow the genus and species names. The title should be re-organized as follows: “…in the blow fly Cochliomyia macellaria (Fabricius) (Diptera: Calliphoridae)”. This should also be changed in the abstract (L. 35).

Throughout the manuscript: Consider using acronyms for the phrases “differentially expressed” and “significantly differentially expressed”. Maybe DE and SDE? Though the precise terminology is appreciated, these phrases make dense results even more difficult to read.

L. 30: Add “of” between “knowledge” and “their”.

L. 47: Consider re-phrasing to “Most research in this realm” or something similar.

L. 61: Consider re-phrasing to “… are used to estimate the minimum PMI/portions of the PMI”.

L. 64 – 65: Consider re-phrasing to “… age-size correlation become increasingly inconsistent” or something similar.

L. 64 – 65: Please include a reference.

L. 67: Please include author name for the first mention of Phormia regina.

L. 106: Change “perform” to “performed”.

L. 107: Add author name as this is the first mention of C. macellaria in the main text of the manuscript.

L. 130: Consider changing “beef” to “bovine”.

L. 132 – 133: Was the amount of beef liver and density of larvae consistent across replicates?

L. 156: Is “UA” supposed to be “USA”?

L. 275: Please italicize C. macellaria.

L. 279 – 281: Please italicize all mentions of Bos taurus and B. taurus.

L. 284: Consider re-phrasing to “… flies are easily visually differentiated as either larval or intrapuparial…”.

L. 291: Please italicize C. macellaria.

Figure 3: If possible, consider coding either “Control” or “Slow” as a unique shape other than a triangle. The duplicate shape, regardless of orientation, makes interpreting the PC plots confusing.

L. 333 and Figure 4 caption: Predication or prediction?

L. 388 – 392: “There did not seem to be an effect of selection regime” à what about the “fast” strain in Figure 4B? This strain seems to have performed much better than the others as the predicted values seem very close to the actual values.

L. 439: Please italicize Bos taurus.

L. 445: Is this the correct figure being referenced (S4)?

L. 456 – 458:  Not sure this is a selling point for your method. Many FE practitioners would argue that a simple visual inspection of the third instar larvae’s crop would suffice for feeding vs. post-feeding differentiation.

L. 466 – 471: Consider listing these other fly species (C. hominivorax, L. sericata, S. calcitrans) in the main text.

Author Response

This is a very thorough and well-executed study investigating the possibility of using microRNAs to age forensically relevant insects. Specifically, the authors focused on the post-feeding third instar larval stages and various timepoints throughout the intrapuparial period, which are all notoriously difficult to accurately estimate using thermal summation models. This work is the culmination of a massive amount of lab experiments and molecular analyses, all of which resulted in some very interesting and useful findings. The authors have also outlined additional future experimentation and analyses that they see as being integral to fully understanding the application of microRNAs in forensic entomology.

A major concern with this and other cutting-edge research concerning novel methods for age estimation is the expense and energy that would need to be invested by an investigating agency to get results that might be only slightly more helpful than using the standard methods (that are much cheaper) that have been around for decades. Don't get me wrong, the research is amazing, but the application to a real death investigation is uncertain. It would greatly strengthen the applied potential of this paper to have a direct comparison of this method to standard estimation methods. How well does the microRNA method hold up to thermal summation modeling? At the very least, a discussion of the importance of such validation work in future studies would be appreciated.

Thank you for these comments and the thorough review.  We have added a statement in the discussion citing the need for future work validating the miRNA method of estimation in comparison to traditional thermal summation models. We agree it is critical to know if it is worth the energy and expense required to perform these additional tests.

Minor Comments:

Title and Abstract: The author name (Fabricius) should directly follow the genus and species names. The title should be re-organized as follows: “…in the blow fly Cochliomyia macellaria (Fabricius) (Diptera: Calliphoridae)”. This should also be changed in the abstract (L. 35).

These items have been changed.  Thank you.

Throughout the manuscript: Consider using acronyms for the phrases “differentially expressed” and “significantly differentially expressed”. Maybe DE and SDE? Though the precise terminology is appreciated, these phrases make dense results even more difficult to read.

Thank you for this suggestion.  First instances of SDE and DE were spelled out (both in text and figure legends). All others were abbreviated.

  1. 30: Add “of” between “knowledge” and “their”.

Done

  1. 47: Consider re-phrasing to “Most research in this realm” or something similar.

Thank you for this suggestion.  Added.

  1. 61: Consider re-phrasing to “… are used to estimate the minimum PMI/portions of the PMI”.

We have added minimum PMI to this statement.  Thank you for the suggestion.

  1. 64 – 65: Consider re-phrasing to “… age-size correlation become increasingly inconsistent” or something similar.

Modified. Thank you for this suggestion

  1. 64 – 65: Please include a reference.

We have added a reference here to Tarone and Foran 2008, which discusses the variability in size in the postfeeding third instar sizes.

  1. 67: Please include author name for the first mention of Phormia regina.

Added.

  1. 106: Change “perform” to “performed”.

Changed.

  1. 107: Add author name as this is the first mention of C. macellaria in the main text of the manuscript.

Added.

  1. 130: Consider changing “beef” to “bovine”.

All instances of “beef” changed to “bovine”

  1. 132 – 133: Was the amount of beef liver and density of larvae consistent across replicates?

A statement was added in the methods to include mass of liver and number of eggs.

  1. 156: Is “UA” supposed to be “USA”?

Corrected

  1. 275: Please italicize C. macellaria.

Corrected. All instances italicized.

  1. 279 – 281: Please italicize all mentions of Bos taurus and B. taurus.

Corrected. All instances italicized.

  1. 284: Consider re-phrasing to “… flies are easily visually differentiated as either larval or intrapuparial…”.

Modified.  Thank you.

  1. 291: Please italicize C. macellaria.

Corrected. All instances italicized.

Figure 3: If possible, consider coding either “Control” or “Slow” as a unique shape other than a triangle. The duplicate shape, regardless of orientation, makes interpreting the PC plots confusing.

These were changed to a diamond shape, while similar to the square shape, we hope it increases the readability of the plot. Thank you for this suggestion

  1. 333 and Figure 4 caption: Predication or prediction?

Predicted.  Thank you for the catch.

  1. 388 – 392: “There did not seem to be an effect of selection regime” à what about the “fast” strain in Figure 4B? This strain seems to have performed much better than the others as the predicted values seem very close to the actual values.

This statement was modified to say: there did not seem to be a large effect of selection regime, suggesting these markers are more ro-bust to the genotypic variation in these selected lines. These results suggest that fast selected lines were more accurately predicted.

  1. 439: Please italicize Bos taurus.

Corrected. All instances italicized.

  1. 445: Is this the correct figure being referenced (S4)?

Thank you for this catch!  This should have been figure 3

  1. 456 – 458:  Not sure this is a selling point for your method. Many FE practitioners would argue that a simple visual inspection of the third instar larvae’s crop would suffice for feeding vs. post-feeding differentiation.

Thank you for this comment. A clause was added to suggest that this would aid in that distinction.  A PCR assay may be more sensitive in its detection than what we can visually identify, but that distinction doesn’t necessarily provide any additional resolution.

  1. 466 – 471: Consider listing these other fly species (C. hominivorax, L. sericata, S. calcitrans) in the main text.

Species names were listed (italicized with name of describer)

Reviewer 2 Report

Whilst, as the authors acknowledge, this is preliminary study with plenty of further work needed to validate this approach this is nonetheless an interesting approach to larval/pupal aging showing the potential for miRNAs in this context. A little proofreading needed (do check the use of italics for species names) and it might be useful to put this analysis in context with it's potential in the real world, in particular regarding the implications if over-or under estimating the development proportion indicated by the miRNA analysis (although further validation studies would address this). overall a well thought out experiment and well written paper.

Author Response

Whilst, as the authors acknowledge, this is preliminary study with plenty of further work needed to validate this approach this is nonetheless an interesting approach to larval/pupal aging showing the potential for miRNAs in this context. A little proofreading needed (do check the use of italics for species names) and it might be useful to put this analysis in context with it's potential in the real world, in particular regarding the implications if over-or under estimating the development proportion indicated by the miRNA analysis (although further validation studies would address this). overall a well thought out experiment and well written paper.

Thank you for your kind words and review of this manuscript.  Species names have all been abbreviated, as well as check of other grammatical issues. A statement was added in the discussion that future validation work should compare estimations of development proportion via miRNA versus traditional thermal summation models in order to test for effectiveness and efficiency.

Reviewer 3 Report

The work is interesting and have scientific significance, but I still have some comments for authors.

1.       I wondered if in order to identify the markers of age why did not the long-lived adults be used?

2.       In line 137-149, larval number of animals in each sample from different stages collected for RNA isolation was not mentioned. What dose “Whole jars were sampled” mean?

3.       The reads in early intrapuparial were much less than others, why? Why there was no error bar for reads from the four replicates?

4.       The symbol of statistical analysis should be given out in some figures but not only described in the figure legends.

Author Response

The work is interesting and have scientific significance, but I still have some comments for authors.

  1. I wondered if in order to identify the markers of age why did not the long-lived adults be used?

Thank you for this comment. To clarify, only immature flies get used for estimations of time of colonization and minimum PMI. As immature are limited in their mobility and dispersion capabilities, they are expected to have been located on the food source for the entirety of their lives (to that point).  Adult flies could have developed at other times and locations and therefore do not provide information about time of colonization.

  1. In line 137-149, larval number of animals in each sample from different stages collected for RNA isolation was not mentioned. What dose “Whole jars were sampled” mean?

A clarifying statement was put into the methods about “whole jars”.  This means that all the individuals from these jars were collected for prospective sequencing or validation work. Additional statements were added in the extraction protocol emphasizing when individuals were used.

  1. The reads in early intrapuparial were much less than others, why? Why there was no error bar for reads from the four replicates?

That is a good question.  Some samples provide lower amount of material than others. All analyses used normalized expression data, so that issues of raw counts did not impact the findings.  We are not sure which error bars are being referred to.  In some plots (Figure 1 and 5 for example) have such little variation in the expression information that the error bars are not visible.  They are there, though.

  1. The symbol of statistical analysis should be given out in some figures but not only described in the figure legends.

Thank you for this suggestion. Letters were added to the boxplots for figure 5 to differentiate those that are statistically different. Shared letters indicate no statistical differences according to Tukey HSD.